# Evaluation of Tire Footprint in Soil Using an Innovative 3D Scanning Method

Weronika Ptak * , Jarosław Czarnecki, Marek Brennensthul , Krzysztof Lejman and Agata Małecka

Institute of Agricultural Engineering, Wroclaw University of Environmental and Life Sciences, 37a Józefa Chełmońskiego Street, 51-630 Wrocław, Poland

* Correspondence: weronika.ptak@upwr.edu.pl

**Abstract:** This paper presents the results of the measurement of tire footprints in soil. The research was conducted under laboratory conditions using soil-filled cases. The research objects were two tires: a radial tire and a bias-ply tire of the same size. The variable parameters were vertical load (7.8 kN, 15.7 kN, 23.5 kN) and inflation pressure (0.8 bar, 1.6 bar, 2.4 bar). Test benches with a mounted tire, a soil case, and a 3D scanner were used in the research. Using the test bench, a tire was loaded with each inflation pressure, and a tire footprint was generated in the soil. Then, a 3D scanner was used to scan the tire footprint, and the parameters of length, width, depth, and tire–soil contact area (as a spatial image) were evaluated using special software. Then, mathematical models were formulated (separately for each type of tire) to describe the tire–soil contact area of the tire footprint as a function of the vertical load and inflation pressure. It was found that the depth of the tire footprint is an important parameter that influences the tire–soil contact area value. However, it was also found that with the right combination of inflation pressure and vertical load, a longer and wider, but shallower, tire footprint can be generated, the contact area value of which is similar to that of a deeper footprint.

**Keywords:** radial tire; bias-ply tire; soil deformation; tire footprint; tire–soil contact area; 3D scanning



## 1. Introduction

In recent years, a continuous increase in the technical efficiency of agriculture has been observed, thanks to which soil tillage can be carried out more efficiently. Different tillage treatments are often combined and performed with complex agricultural machinery with larger operating widths, and thus a larger mass. Such machinery generates pressure on the soil and compacts it through wheel traffic, which leads to its degradation [1]. In consequence, the water–air balance of the soil is disturbed, the soil's capacity to absorb rainwater is diminished, and it is insufficiently ventilated, which results in erosion [2–6]. Excessive soil compaction can reduce the yield of plants, as their root system, which is responsible, among other things, for the uptake of water and nutrients from the soil, is not developed sufficiently [7].

The effects of soil compaction are far-reaching, and solutions to mitigate its consequences are being sought. Growing deep-rooted plants might help, as this contributes to soil loosening [8–11]. In agricultural practice, the size and weight of the chassis and wheels of machines used during tillage treatments are important. A typical agricultural machine is based on a wheeled system, the essential element of which is the tire, which is in direct contact with the soil. According to their internal structure, tires can be divided into radial and bias-ply tires. The former is manufactured with an additional layer of material to reinforce the tread part, while the side parts are flexible. On the other hand, the main feature of a bias-ply tire is its greater stiffness, which is due to the same amount of material being evenly spread throughout its cross-section [12]. In practice, this means that a bias-ply tire is more resistant to mechanical damage, but due to its greater stiffness, it can have a more destructive effect on the ground, compacting the soil more than a radial tire.

Due to its having direct contact with the soil, a tire is responsible for the amount of pressure generated by agricultural machines, and the pressure value depends on the tire–soil contact area. This is one of the key tire footprint parameters used to compare the effects of different kinds of tires and the different conditions they operate in [13,14]. Factors that affect the tire–soil contact area include the stiffness of the tire, its size, its inflation pressure, and the vertical load it is subjected to during movement [15]. The literature offers many studies that measured tire contact area with the soil under the influence of variable factors. On the basis of measurements, Grečenko [16] presented formulas predicting the tire–soil contact area, while other authors [17–19] described the tire footprint on the soil as being in the shape of a super-ellipse and included, among other aspects, its length and width in the formula. Technological progress has made it possible to study tire footprints using digital image analysis [20–22]. Kenarsari et al. [23] used photogrammetry to create a 3D model of a tire footprint and then analyzed its length, width, and volume. Farhadi et al. [24] created plaster of Paris molds of a footprint and then used a 3D scanner to obtain information about its dimensions.

According to the literature, many factors, such as tire internal structure, vertical load, and inflation pressure, affect tire contact area with the soil. The area is a very important parameter for determining the distribution of forces applied to the soil. In order to minimize soil compaction, it is advisable to constantly obtain information on the shape and dimensions of the tire footprint in the soil. Many studies analyzed the contact area as a flat surface. However, the tire side edges also interact with the soil, creating a spatial footprint. Today's level of technology makes it possible to obtain information about a three dimensional tire footprint and facilitates a more accurate analysis of the results. Taking into account the above, the aim of the present study was to assesses the impact of radial and bias-ply tires subjected to selected conditions on the shape of tire footprints in soil using 3D scanning techniques and digital image analysis.

## 2. Materials and Methods

This research was conducted under laboratory conditions using sandy loam soil. Its moisture and compactness were kept constant throughout, at 25% and 0.9 MPa, respectively. Both were measured with a Penetrologger set produced by Eijkelkamp. The compactness of the soil was measured with a cone, which was part of the set, with a top angle of $60°$, a base area of 0.0001 $m^2$, and a penetration velocity of 3 cm·$s^{-1}$. The soil moisture was measured with a ThetaProbe, which was also included in the set. Two agricultural tires of the same size but with different structural types, radial and bias-ply, were tested (profile width: 500 mm, profile height: 250 mm, and rim diameter: 17 inches). As part of the tests, three inflation pressure (p) levels were used (0.8 bar, 1.6 bar, and 2.4 bar), with three values of vertical load (G) acting on the tire: 7.8 kN, 15.7 kN, and 23.5 kN. The research included the measurement of the length (l), width (b), and depth (h) of the tire footprint in the soil and the tire–soil contact area ($A_s$). The research was conducted using the methodology of Ptak et al. [25] and Ptak et al. [26]. Unlike in the abovementioned research, a tire footprint in the soil was scanned, so the test bench and the scanning process required modification.

### 2.1. Test Bench

To generate a tire footprint in the soil, a unique test bench was used (Figure 1). Its design allowed for a smooth change in vertical load, and at each stage it was also possible to change the tire inflation pressure. The removable part of the test bench was a soil-filled case (1) with a length of 1000 mm, a width of 1000 mm, and a height of 600 mm. Between the outer frame (3) and the inner frame (4), a hydraulic jack (6) was mounted in the vertical plane. The vertical load was smoothly changed with the jack, and its value was measured with a TecSis inductive dynamometer (5), with a accuracy of 50 N and a measuring range of between 0 and 100 kN. The tire was mounted on a shaft with bearings, with the former attached to the inner frame (4). The screw mechanisms (7) allowed for the locking of the

inner frame and the prevention of its movement and pressure drops in the hydraulic jack, which would result in an unintended reduction in the vertical load.

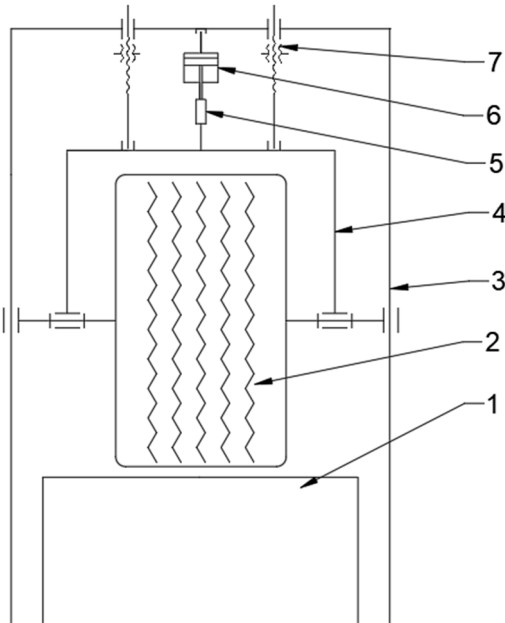

**Figure 1.** The test bench: 1—soil case; 2—wheel with tire; 3—main frame; 4—inner frame; 5—dynamometer; 6—hydraulic cylinder; 7—screw mechanism for locking the inner frame.

The research was conducted in static conditions, so the wheel was placed in the soil case and the tire footprint was generated in the soil without applying torque. There were no driving or braking forces, which could affect the pressure distribution in the soil. After making a footprint in the soil, the soil case was pulled off of the bench. Then, markers indicating the edges were placed around the footprint, which allowed for a more accurate line to be drawn between the footprint and the rest of the soil surface. Before each footprint was created, the soil was mechanically loosened and then compacted to the previous value.

### 2.2. Scanning Process

The tire footprint in the soil was scanned with a 3D scanner (SMARTTECH3D Universe), the technical specifications of which are presented in Table 1. The scanner was connected to a laptop with special SMARTTECH3D measuring software, which allows for continuous observation of the acquired data.

**Table 1.** 3D scanner specifications.

| Parameter | Description |
| --- | --- |
| Scanning technology | white structural light—LED |
| Measuring volume (x × y × z) (mm) | 400 × 300 × 240 |
| Distance between points (mm) | 0.156 |
| Accuracy (mm) | 0.08 |
| Power consumption during measurement (W) | 200 |
| Mass (kg) | 4.40 |
| Working temperature (°C) | 20 ± 0.5 |

The 3D scanner and laptop (Figure 2) were mounted on a tripod column, which made it possible to maintain a constant height of the scanner's position over the scanned footprint. This helped facilitate the preservation of the measuring volume of the scanner and the efficient movement of the measuring device around the soil case. As a result of the scan, a point cloud was obtained that reproduced the shape and geometry of the tire footprint in

the soil. However, in order to facilitate a proper analysis, it was necessary to first remove the scan of the soil outside the tire footprint, which would disturb the measurement results (for this reason, it was necessary to use the markers mentioned above), and then create a mesh of triangles built from the points.

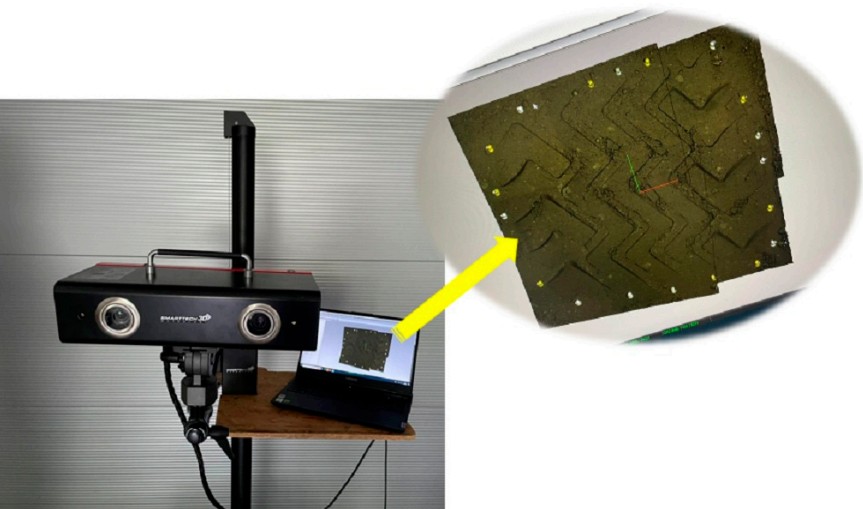

**Figure 2.** 3D scanner and the resulting point cloud.

Figure 3 shows a mesh of triangles of a tire footprint in the soil, with the length and width dimensions taken during the measurement. The tire–soil contact area ($A_s$) was available in the SMARTTECH3D measure software as the whole footprint in three-dimensional space. Scientists usually measure the tire–soil contact area using a simplification, in the form of a two-dimensional projection area. In the case of the presented technique, this a novel approach that makes it possible to present the real shape and size of tire footprints.

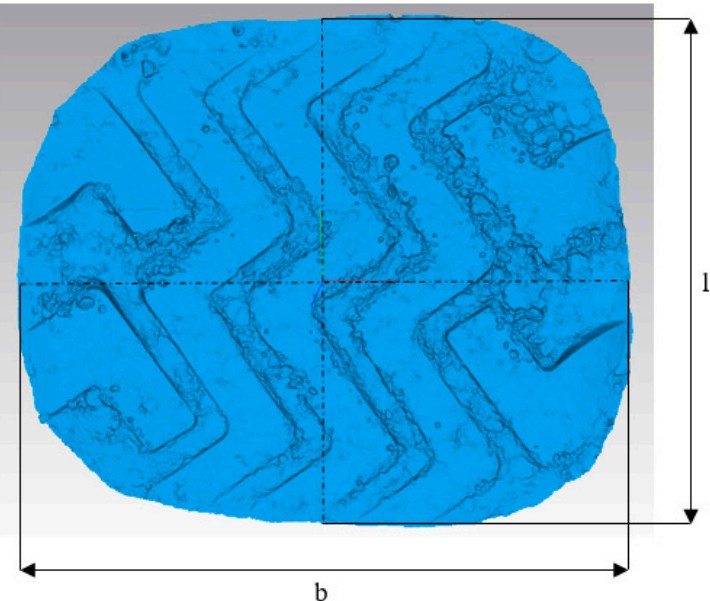

**Figure 3.** Mesh of triangles of a tire footprint in soil. l—length; b—width.

In order to measure the depth of the footprint (h), it was necessary to create its vertical cross-section (Figure 4). Footprint parameters such as length, width, and depth were always measured across the middle of the footprint.

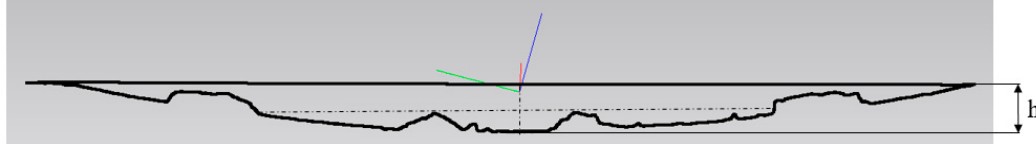

**Figure 4.** Vertical section of the tire footprint in the soil. h—depth of the footprint.

*2.3. Statistical Analysis*

For verification purposes, Statistica 12.5 (StatSoft) was used to perform statistical analyses of the results. As part of this analysis, a homogeneity of variance test was performed, and the compatibility of the data with the normal distribution was assessed. Next, a two-factor analysis of variance at a significance level of $\alpha = 0.05$ was performed, together with an analysis of the homogeneous groups, as part of a post hoc test. The next step in the statistical analysis was the development of mathematical models describing the footprint area as a function of the operating parameters (vertical load and tire inflation pressure). The model was developed using TableCurve and was verified through standard calculations using an Excel spreadsheet.

## 3. Results

Figure 5 shows the length values of footprints generated by radial and bias-ply tires. The length was the smallest (328 mm for the radial tire and 421 mm for the bias-ply tire) when a vertical load of 7.8 kN was applied at an inflation pressure of 2.4 bar for the radial tire and 1.6 bar for the bias-ply tire. In most cases, with the same load and pressure values, the footprint length was greater for the bias-ply tires. For both tires, the footprint length increased with an increase in the vertical load (with the exception of the bias-ply tire when the vertical load increased to 15.7 kN at an inflation pressure of 1.6 bar, when it decreased by about 4 mm).

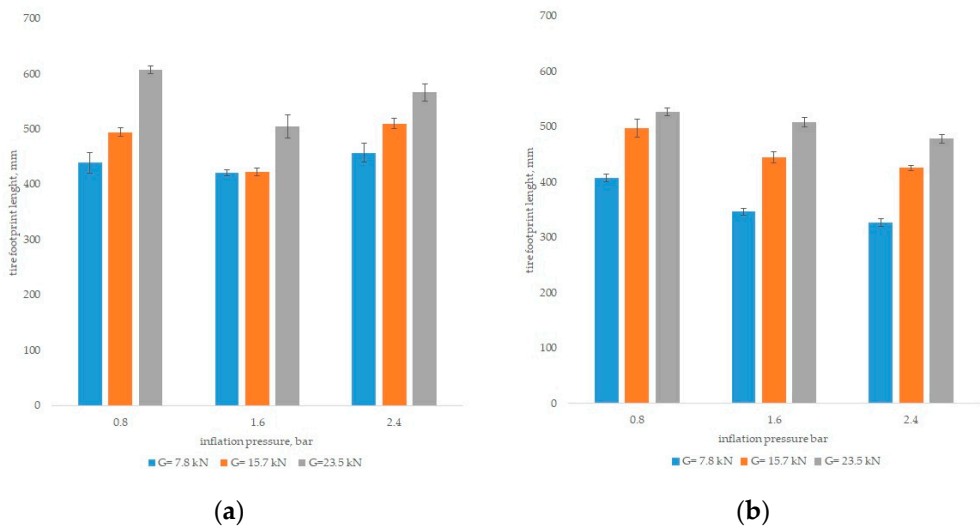

| (a) | (b) |

**Figure 5.** Footprint length values: bias-ply tire (**a**); radial tire (**b**); the linear markers represent the standard deviation.

For the bias-ply tire, a larger increase in the footprint length was observed when the vertical load was increased from 15.7 kN to 23.5 kN than when it was increased from 7.8 kN to 15.7 kN. In the former case, it increased by 23%, 20%, and 11% for inflation pressures of 0.8 bar, 1.6 bar, and 2.4 bar, respectively. However, after the first increase in load (from 7.8 kN to 15.7 kN), the largest length increase for the bias-ply tire was only 13%. The opposite trend was observed for the radial tire; in this case, the first load increase resulted in a greater increase in footprint length, by 22%, 28%, and 30% for inflation pressures of

0.8 bar, 1.6 bar, and 2.4 bar, respectively, while with a vertical load of 23.5 kN, the maximum increase was only 14%. Unlike for the bias-ply tire, for the radial tire, reducing the pressure always resulted in an increase in the footprint length. Values that were 4–17% larger were recorded when the pressure was decreased from 1.6 bar to 0.8 bar; for the first load increase, the length increase was greater than for the second. In the case of the bias-ply tire, an increase was observed only after the pressure was reduced from 1.6 bar to 0.8 bar (by 4%, 7%, and 20% for vertical loads of 7.8 kN, 15.7 kN, and 23.5 kN, respectively). The pressure reduction from the highest to the middle value resulted in a decrease in the length of the footprint by 7–11%, depending on the vertical load.

Another footprint parameter studied in our research was width (Figure 6). Its highest value was 502 mm for a radial tire with a vertical load of 23.5 kN and an inflation pressure of 0.8 bar. With the same load and pressure, the footprint width for the bias-ply tire was 497 mm. This difference might have been due to the fact that radial tires are only more susceptible to lateral deformation at low inflation pressures. As with the length of the radial tire footprint, its width increased with a reduction in the tire inflation pressure at a given vertical load; after both the first and second pressure reduction, the average increase in the width of the footprint was 4%. For the bias-ply tire, on the other hand, the increase in the footprint width due to the pressure reduction was more pronounced between the levels of 1.6 bar and 0.8 bar (an increase of 2%, 10%, and 7% for loads of 7.8 kN, 15.7 kN, and 23.5 kN, respectively). After reducing the pressure from 2.4 bar to 1.6 bar, an increase in the width of the generated footprint was only found at the lowest vertical load (an increase of 2%). At a load of 15.7 kN, the same pressure drop resulted in a 5% reduction in the width of the footprint, while at the highest vertical load, pressure reduction did not result in any changes.

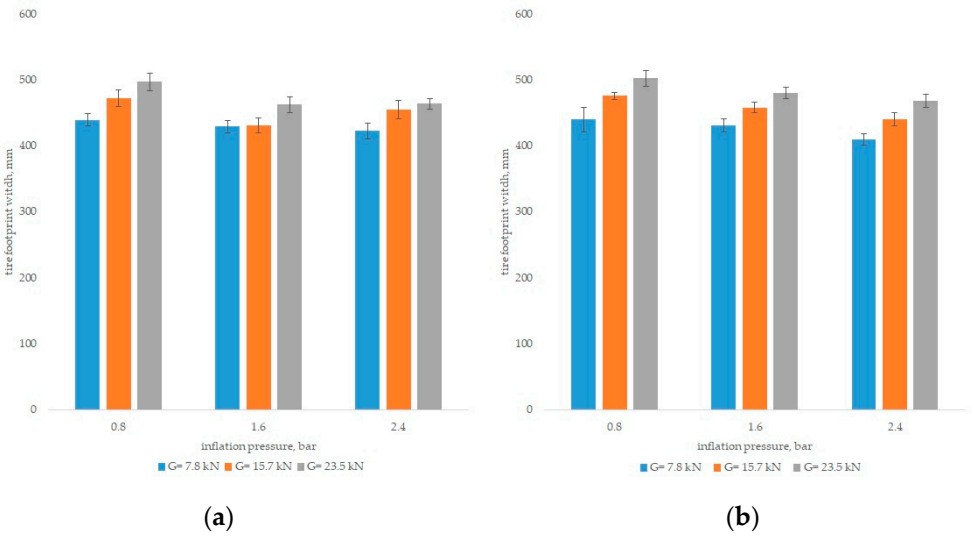

(a)                                                    (b)

**Figure 6.** Footprint width values: bias-ply tire (**a**); radial tire (**b**); the linear markers represent the standard deviation.

Figure 7 presents the depth values of the footprints in the soil for the radial and bias-ply tires. Noticeably, the highest (54 mm) value was observed for the bias-ply tire with an inflation pressure of 2.4 bar and a vertical load of 23.5 kN. With the highest inflation pressure (2.4 bar) for the bias-ply tire, as the vertical load increased, the depth of the footprint also increased, while at lower inflation pressure values (0.8 bar and 1.6 bar), an increase in the vertical load from 7.8 kN to 15.7 kN resulted in a decrease in the depth of the footprint, but it increased again with a vertical load of 23.5 kN. The lowest value of footprint depth of 15 mm for the bias-ply tire was recorded with a vertical load of 15.7 kN and an inflation pressure of 0.8 bar. In addition, for the bias-ply tire, at the highest inflation pressure (2.4 bar), both increases in the vertical load of the tire resulted in an increase in

the footprint depth (increases of 30% and 43%, respectively). At the middle pressure value (1.6 bar), the first increase in vertical load resulted in an 11% decrease in the depth of the footprint, while the next increase resulted in a 10% increase. At the lowest inflation pressure, with a load of 7.8 kN and 15.7 kN for the bias-ply tire, the depth values were practically the same, while after increasing the vertical load to 23.5 kN, an increase of 68% was observed. In the case of the radial tire, the highest depth value was 26 mm (with a load of 23.5 kN and an inflation pressure of 2.4 bar), only slightly different from that produced by 15.7 kN. For each inflation pressure value, an increase in the vertical load resulted in an increase in the depth of the tire footprint. The largest difference of 14 mm (108%) was observed at a pressure of 2.4 bar, between the lowest and highest vertical load. At the two lower inflation pressures, the differences in the depths of the footprints (between sequential load levels) were in the range of 19–39%. For both tires, a clear effect of the inflation pressure on the depth of the generated footprint was noted. In most cases, a pressure drop resulted in a reduction in the depth, while for the bias-ply tire the average reductions in the depth due to the first and second pressure drops were at a similar level (about 30%), whereas for the radial tire, only the pressure reduction from 1.6 bar to 0.8 bar resulted in shallower footprints. On the other hand, at the highest inflation pressure, the radial tire produced a footprint more than twice as shallow as the bias-ply tire.

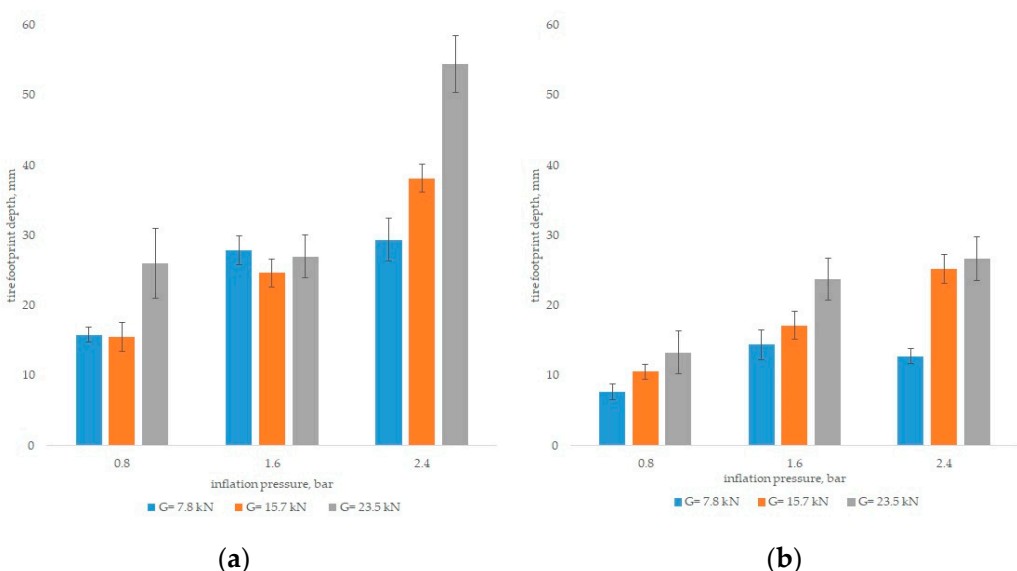

**Figure 7.** Footprint depth values: bias-ply tire (**a**); radial tire (**b**); the linear markers represent the standard deviation.

The last footprint parameter was the tire–soil contact area (Figure 8). Its highest value (0.33 m$^2$) was observed for the bias-ply tire at the lowest pressure and the highest vertical load. For the bias-ply tire, at the same vertical loads, the lowest value of the footprint area was noted at an inflation pressure of 1.6 bar. However, at each level of inflation pressure, an increase in the vertical load resulted in an increase in the contact area. The highest increases were found at the first change in inflation pressure (by 43% and 23% for increases in load from 7.8 kN to 15.7 kN and from 15.7 kN to 23.5 kN, respectively).

For the radial tire, the lowest contact area of 0.13 m$^2$ was found with a vertical load of 7.8 kN and an inflation pressure of 2.4 bar. It was also observed that the footprint area for the radial tire changed according to a certain trend; an increase in the vertical load at a given inflation pressure resulted in an increase in the contact area, and at each inflation pressure, the first increase in load (from 7.8 kN to 15.7 kN) resulted in a larger area increase than the second, by 44%, 47%, and 62% for inflation pressures of 0.8, 1.6, and 2.4 bar, respectively. At the same time, a reduction in the inflation pressure in the radial tire at a given vertical load resulted in an increase in the contact area; the largest differences were found for the first

load, with the area increasing by 20% after reducing the pressure from 1.6 bar to 0.8 bar, and by 15% after reducing from 2.4 bar to 1.6 bar. For the first load, reducing the inflation pressure from the highest to the lowest value resulted in an increase in the contact area by 38% (to 0.18 m$^2$).

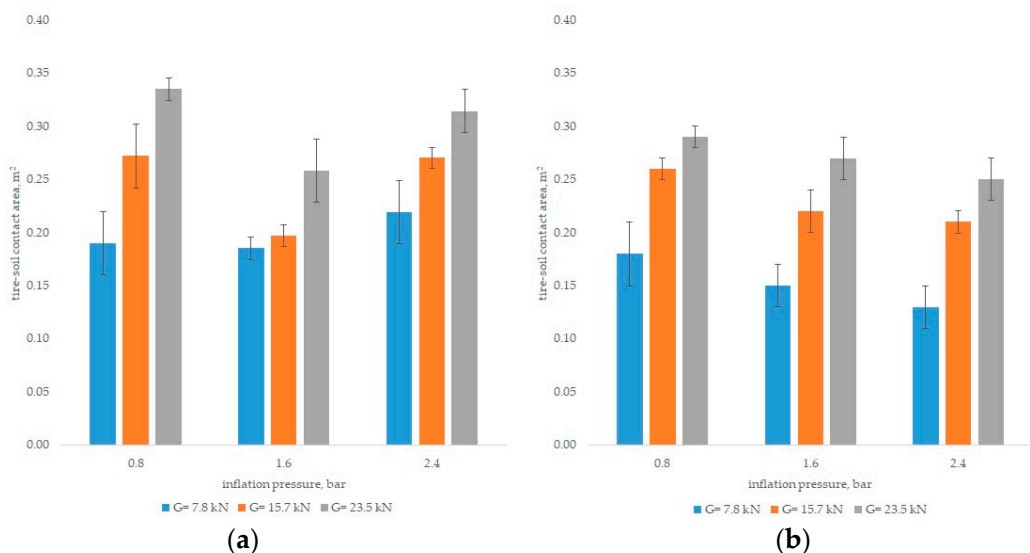

**Figure 8.** Tire–soil contact area values: bias-ply tire (**a**); radial tire (**b**); linear markers represent the standard deviation.

Table 2 presents a statistical analysis of the experimental data for the radial tire. The footprint parameters are the width, length, depth, and tire–soil contact area. The *p*-values presented in the table indicate the probability of accepting the hypothesis that the factor does not affect the imprint parameter. If the value of p does not exceed the significance level $\alpha$ (0.05), the factor had a significant influence on the analyzed parameter.

**Table 2.** Statistical analysis of the experimental data for the radial tire. Significance level $\alpha$ = 0.05. SD—standard deviation.

| Footprint Parameter | Factor | Factor Level | Arithmetic Mean | $\pm$SD | *p*-Value |
|---|---|---|---|---|---|
| Width of the footprint (b), mm | Vertical load | 7.8 kN | 426.8 [A] | 15.1 | <0.0001 |
| | | 15.7 kN | 458.1 [B] | 16.6 | |
| | | 23.5 kN | 483.4 [C] | 17.5 | |
| | Inflation pressure | 0.8 bar | 472.6 [A] | 29.5 | <0.0001 |
| | | 1.6 bar | 456.2 [B] | 22.8 | |
| | | 2.4 bar | 439.6 [C] | 22.4 | |
| Length of the footprint (l), mm | Vertical load | 7.8 kN | 360.6 [A] | 36.5 | <0.0001 |
| | | 15.7 kN | 456.4 [B] | 33.7 | |
| | | 23.5 kN | 504.8 [C] | 22.5 | |
| | Inflation pressure | 0.8 bar | 477.5 [A] | 55.1 | <0.0001 |
| | | 1.6 bar | 433.6 [B] | 70.8 | |
| | | 2.4 bar | 410.7 [C] | 66.6 | |
| Depth of the footprint (h), mm | Vertical load | 7.8 kN | 11.6 [A] | 3.3 | <0.0001 |
| | | 15.7 kN | 17.6 [B] | 6.5 | |
| | | 23.5 kN | 24.2 [C] | 6.6 | |
| | Inflation pressure | 0.8 bar | 10.5 [A] | 3.0 | <0.0001 |
| | | 1.6 bar | 18.4 [B] | 4.7 | |
| | | 2.4 bar | 21.6 [C] | 6.9 | |

**Table 2.** *Cont.*

| Footprint Parameter | Factor | Factor Level | Arithmetic Mean | ±SD | *p*-Value |
|---|---|---|---|---|---|
| Tire–soil contact area ($A_s$), m$^2$ | Vertical load | 7.8 kN | 0.154 $^A$ | 0.029 | <0.0001 |
| | | 15.7 kN | 0.229 $^B$ | 0.026 | |
| | | 23.5 kN | 0.271 $^C$ | 0.023 | |
| | Inflation pressure | 0.8 bar | 0.244 $^A$ | 0.053 | <0.0001 |
| | | 1.6 bar | 0.214 $^B$ | 0.055 | |
| | | 2.4 bar | 0.197 $^B$ | 0.055 | |

The letters in the arithmetic mean column (A, B, C) denote separate homogenous groups.

Based on the footprint generated by the radial tire, it can be concluded that both experimental factors (vertical load and tire inflation pressure) had a significant influence on all footprint dimension values (the *p*-values were much lower than the assumed significance level $\alpha$). In the case of the first factor (vertical load), each of its levels formed a separate homogeneous group; these trends were observed for all four footprint parameters. For the second factor (tire inflation pressure), separate homogeneous groups were identified for each level for the length, width, and depth of the footprint, while for the fourth parameter (contact area), two homogeneous groups were obtained; the first for the lowest tire pressure level (0.8 bar), and the second for the other two levels (1.6 bar and 2.4 bar). In practice, this meant that the change in inflation pressure from 2.4 bar to 1.6 bar did not result in significant changes in the tire–soil contact area.

Table 3 presents the results of the statistical analysis in relation to the footprints generated by the bias-ply tire. Both the experimental factors and the footprint parameters were the same as in the case of the radial tire. First, the effect of the experimental factors on the footprint parameters was determined; in all cases, it was found that both the inflation pressure and vertical load had a significant impact on the footprint parameters. Subsequently, homogeneous group tests were performed. When analyzing the impact of the vertical load, it turned out that a change in its level resulted in significant changes in all footprint parameters, except for the depth; in this case, significant differences were only found between 23.5 kN and the other two levels (there was no significant difference in depth values between 7.8 kN and 15.7 kN). The influence of the tire inflation pressure on the footprint parameters turned out to be slightly smaller; only in the case of the depth of the footprints were three separate homogeneous groups identified, while for the remaining dimensions, two homogeneous groups were observed.

**Table 3.** Statistical analysis of the experimental data for the bias-ply tire. Significance level $\alpha$ = 0.05. SD—standard deviation.

| Footprint Parameters | Factor | Factor Level | Arithmetic Mean | ±SD | *p*-Value |
|---|---|---|---|---|---|
| Width of the footprint (b), mm | Vertical load | 7.8 kN | 430.5 $^A$ | 11.5 | <0.0001 |
| | | 15.7 kN | 452.8 $^B$ | 20.9 | |
| | | 23.5 kN | 474.3 $^C$ | 19.6 | |
| | Inflation pressure | 0.8 bar | 469.6 $^A$ | 26.9 | 0.0002 |
| | | 1.6 bar | 441.1 $^B$ | 18.6 | |
| | | 2.4 bar | 447.0 $^B$ | 21.3 | |
| Length of the footprint (l), mm | Vertical load | 7.8 kN | 439.1 $^A$ | 20.4 | <0.0001 |
| | | 15.7 kN | 475.5 $^B$ | 41.2 | |
| | | 23.5 kN | 559.4 $^C$ | 46.5 | |
| | Inflation pressure | 0.8 bar | 513.6 $^A$ | 74.9 | <0.0001 |
| | | 1.6 bar | 449.4 $^B$ | 43.1 | |
| | | 2.4 bar | 511.1 $^A$ | 48.9 | |

**Table 3.** *Cont.*

| Footprint Parameters | Factor | Factor Level | Arithmetic Mean | ±SD | *p*-Value |
|---|---|---|---|---|---|
| Depth of the footprint (h), mm | Vertical load | 7.8 kN | 24.3 [A] | 6.7 | 0.0004 |
| | | 15.7 kN | 26.1 [A] | 10.0 | |
| | | 23.5 kN | 35.8 [B] | 14.5 | |
| | Inflation pressure | 0.8 bar | 19.1 [A] | 5.8 | <0.0001 |
| | | 1.6 bar | 26.5 [B] | 2.5 | |
| | | 2.4 bar | 40.7 [C] | 11.4 | |
| Tire–soil contact area ($A_s$), m$^2$ | Vertical load | 7.8 kN | 0.199 [A] | 0.027 | <0.0001 |
| | | 15.7 kN | 0.247 [B] | 0.039 | |
| | | 23.5 kN | 0.301 [C] | 0.038 | |
| | Inflation pressure | 0.8 bar | 0.264 [A] | 0.065 | 0.0004 |
| | | 1.6 bar | 0.216 [B] | 0.037 | |
| | | 2.4 bar | 0.267 [A] | 0.043 | |

The letters in the arithmetic mean column (A, B, C) denote separate homogenous groups.

### 3.1. Mathematical Models of Static Soil Deformation

As part of the statistical analysis, mathematical models were developed to describe the contact area as a function of the tire operational parameters. The choice of the contact for the model was dictated by the fact that it is crucial for the value of the force exerted on the soil and for forecasting the risk of soil compaction. Due to the structural differences between the tires, models were developed separately for the radial and bias-ply tires. Mathematical modelling was carried out using Statistica 12.5 (Statsoft) and TableCurve 2D ver. 5.0.1, Systat Software, San Jose, CA, USA.

### 3.1.1. Mathematical Model for Radial Tire Footprint

After verifying the significance of the experimental factors, a mathematical model was developed. First, the normality of the distribution of the variables (footprint area values) was tested. For this purpose, a Shapiro–Wilk test at a significance level of $\alpha = 0.05$ was used. The value of the test function W was 0.94, and the probability *p* value was 0.207, which led to the conclusion that the data had a normal distribution (this is the case when the *p*-value is greater than the assumed significance level $\alpha$). Subsequently, a general form of the mathematical model (Equation (1)) was developed:

$$A_s = -0.267 - 0.0427 \cdot lnp + 0.012 \cdot \left( ln\frac{G}{0.00981} \right)^2 - 0.0035 \cdot \left( \frac{G}{0.00981} \right)^{0.5} \quad (1)$$

where:

$A_s$—contact area of the footprint (m$^2$),
*p*—inflation pressure in the tire generating the footprint (bar),
*G*—vertical load of the tire generating the footprint (kN).

The value of the coefficient of determination $R^2$ for the model was 0.916, and the mean absolute error of estimation was 0.006. As part of the model fit analysis, a significance test of the model variables was performed, formulating a null hypothesis about their insignificance. To verify this hypothesis, a test using the F-Snedecor function at the significance level of $\alpha = 0.05$ was used (Equation (2)):

$$F = \frac{\frac{R^2}{k}}{\frac{(1-R^2)}{n-k-1}} \quad (2)$$

where:

$R^2$—coefficient of determination,
*n*—number of cases,
*k*—number of variables.

If the value of $F$ is higher than the critical value ($F_{crit}$), the null hypothesis regarding the insignificance of the variables in the model is rejected. The $F$ function value calculated on the basis of the above formula was 130.85, and the critical value from the F-Snedecor distribution tables was 4.26; therefore, the null hypothesis was rejected. This means that the variables of the regression model were significant. Subsequently, as part of the model verification, a so-called similarity grid was prepared, i.e., a graph illustrating the relationship between the data calculated from the model and the actual data (Figure 9).

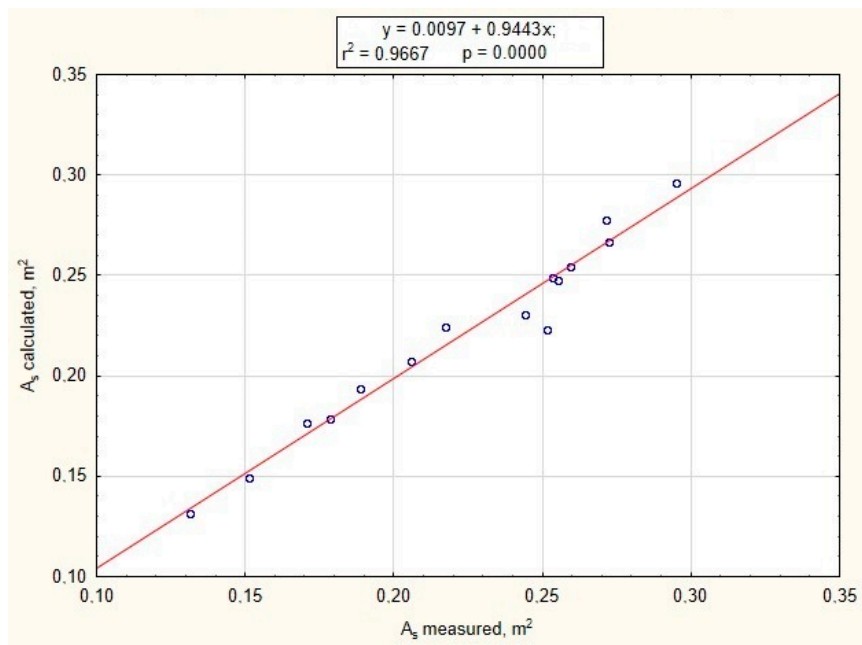

**Figure 9.** Relationship between the model data and the actual data for the radial tire ($A_s$—contact area).

By analyzing the graph presented in Figure 9, it turns out that the discrepancy between the actual data and the data calculated from the model is in most cases small. Only the points corresponding to the values of 0.231 and 0.244 m$^2$ deviate significantly from the trend line. To determine the relationships between the calculated and measured data, evaluation of the regression significance was conducted (F-Fisher test at a significance level of $\alpha = 0.05$ was used). The null hypothesis stated that the regression coefficient and slope were statistically insignificant. In the case of the coefficient of regression, the value of the test function was $F_{(1, 13)} = 377.37$ and the probability of the acceptance of the null hypothesis had a value lower than 0.00001. A relatively high test function value and very low level of probability caused us to reject the null hypothesis; for this reason, the regression coefficient was significant. However, the test procedure for the slope showed that it was insignificant (probability $p = 0.3978$).

### 3.1.2. Mathematical Model for the Bias-Ply Tire Footprint

In the case of the bias-ply tire, a mathematical model was developed in the same way as for the radial tire. The test of the normality of the distribution (Shapiro–Wilk at the significance level of $\alpha = 0.05$) confirmed that the data had a distribution that was consistent with a normal distribution; the value of the test function (W) was 0.94, and the probability level of the rejection of the hypothesis of no normal distribution was $p = 0.185$. For the bias-ply tire, the following mathematical model (Equation (3)) with the contact area as a function of the tire operating parameters was developed:

$$A_s = 0.947 - \frac{1.882}{p^{0.5}} + \frac{1.063}{p} + 7.76 \cdot 10^{-4} G \cdot ln \frac{G}{0.00981} \tag{3}$$

where:

$A_s$—contact area of the footprint (m$^2$),

$p$—inflation pressure in the tire generating the footprint (bar),

$G$—vertical load of the tire generating the footprint (kN).

The value of the coefficient of determination ($R^2$) for the developed model was 0.812, and the mean absolute error of estimation was 0.014.

The correctness of the model fit was verified using the F-Snedecor function. The calculated value of the function was $F = 51.83$, while the critical value in the F-Snedecor distribution tables was 19.45; therefore, it was concluded that the model was well suited to the empirical data. Figure 10 shows the relationship between the model data and the actual data.

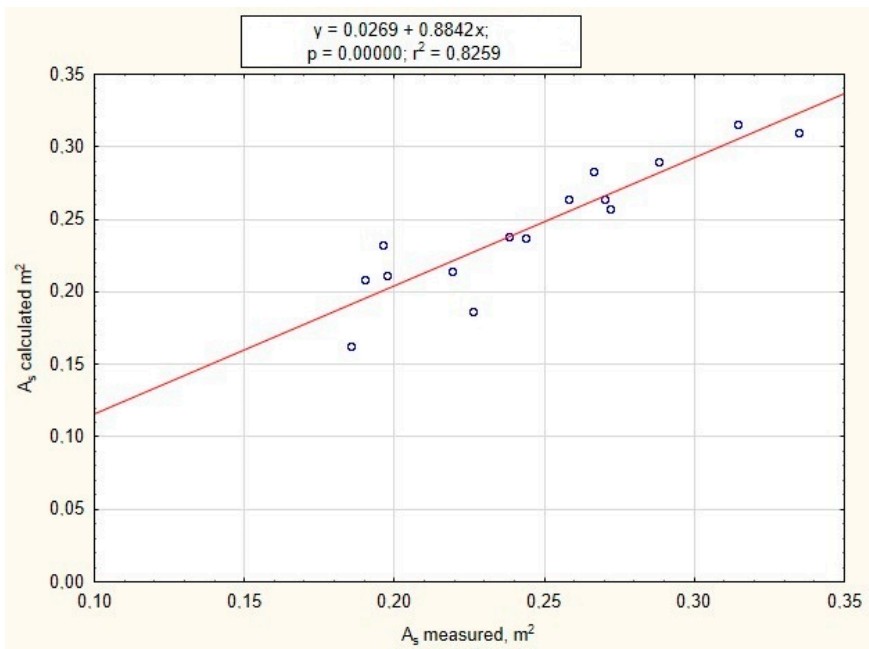

**Figure 10.** Relationship between the model data and the actual data for the bias-ply tire (A$_s$—contact area).

In the case of the bias-ply tire model, a slightly worse fit was observed than in the case of the radial tire. This was also evidenced by the lower value of the coefficient of determination $R^2$ (0.812 for the bias-ply tire model and 0.916 for the radial tire model). When analyzing Figure 10, it can be seen that at higher values of the real area (A$_s$), the match was good, but at lower values, the dispersion of the points from the trend line was large. In the case of the bias-ply tire, verification of the model fit was conducted in the same way as in the case of the radial tire (F-Fisher test at the significance level of $\alpha = 0.05$). The evaluation of the regression coefficient showed that it was significant. The parameters were the values of test function $F_{(1, 13)} = 61.662$, probability level $p < 0.00001$. The slope was an insignificant parameter (probability $p = 0.3573$).

## 4. Discussion

Our study was carried out using soil as a deformable substrate and different values of tire inflation pressure and vertical load. It was possible to observe changes in the dimensions of the soil footprint of tires with different internal constructions but the same external sizes. Footprint parameters such as length, width, and depth were used to determine the contact area of the tire with the soil (without simplifying it, for example, to the cross-sectional area of the tire at its contact with the ground).

According to our results, the tire inflation pressure and vertical load are the factors that affect the dimensions of the tire footprint in the soil. In most cases, a reduction in

the inflation pressure increases the footprint dimensions, which was confirmed by Shao et al. [27], O'Sullivan et al. [28], and Keller [29]. In addition to the length and width of the footprint, an important parameter is its depth, on which the tire–soil contact area largely depends. Hemmat et al. [30] suggested that the depth of the footprint is the main indicator of soil compaction. According to Moitzi et al. [31], a higher value of vertical load at a lower inflation pressure increases the tire footprint depth in the soil, which was confirmed by Rapper et al. [32]. In the present research, in some cases the same tire–soil contact area was observed for different values of tire inflation pressure and vertical load. For example, a contact area of 0.26 m$^2$ was recorded at an inflation pressure of 1.6 bar and 23.5 kN vertical load (bias-ply tire) and 0.8 bar and 15.7 kN (radial tire). However, at the same time, for the bias-ply tire, a much greater footprint depth was recorded than for the radial tire (26.99 mm and 10.56 mm, respectively). It was also noted that the same values of inflation pressure and vertical load (0.8 bar and 7.8 kN) resulted in a similar contact area for both tires, but the footprint depth for the bias-ply tire was 15.83 mm, and for the radial tire it was 7.62 mm. These examples show that, under certain conditions, a radial tire has a less destructive effect on the ground. When comparing radial and bias-ply tires, Kurjenluoma et al. [33] found that their internal structure also affects the formation of the rut (tire footprint in the soil) and that lowering the tire pressure reduces the rut depth, but only on soft soil with high humidity. Farhadi et al. [24] took into account soil moisture as a factor influencing the dimensions of the tire footprint in the soil. Similarly, Mohsenimanesh and Ward [34] noted that increasing the soil moisture causes an increase in the tire–soil contact area, but they also found that at any level of soil moisture, the vertical load of the tire also affects this parameter. Comparing the effects of two radial tires, Schjønning et al. [35] noted that a tire with a smaller width generated a longer footprint and was less sensitive to inflation pressure values not recommended by the manufacturer than a wider tire. Botta et al. [36] noted that, in order to increase the contact area of a tire with soil, the size of the tires, the vertical load on the tires, and the soil moisture should be taken into account.

Based on the literature review, our results, and a comparative analysis with those of other authors, it can be concluded that knowledge of the contact area of a tire with the soil and information on factors affecting its value are crucial for protecting soil against the negative impact of agricultural tires. It should also be noted that it is necessary to take into account all the variables described above at the same time, because their selective analysis may lead to erroneous conclusions (e.g., different load values can result in the same contact area and different depths). The right combination of factors, such as load and pressure, can have a positive impact on soil protection and thus improve crop production results.

## 5. Conclusions

Analysis of our results made it possible to conclude that both research factors, i.e., vertical load on the tire and tire inflation pressure, had an impact on the footprint dimension values:

1.  An increase in the vertical load, at the same tire inflation, resulted in an increase in the length of the tire footprint. For the radial tire, the length increased steadily with the same load and a decrease in inflation pressure. In the case of the bias-ply tire, the length decreased when the pressure dropped from 2.4 bar to 1.6 bar and then increased when it dropped from 1.6 bar to 0.8 bar. In most cases, at the same pressure and load, greater tire footprint lengths were observed for the bias-ply tire than for the radial tire.
2.  At the same inflation pressure, when the vertical load was increased, the width of the footprint also increased for both the radial and bias-ply tires. At the same time, a reduction in the inflation pressure with the same vertical load resulted in an increase in the width of the tire footprint, but only for the radial tire. With the same values of vertical load and inflation pressure, in most cases the radial tire imprint was wider.
3.  Reducing the vertical load on the tires resulted in a decrease in the depth of the footprint for all inflation pressure values, but only for the radial tire (for the bias-ply tire, this trend was observed only when the pressure rose to its highest value). In all

cases, at the same values of inflation pressure and vertical load, significantly higher tire footprint depths were observed for the bias-ply tire.

4. Increasing the vertical load at a constant inflation pressure caused an increase in the contact area with the soil of the tested tire footprint. For the radial tire, a reduction in its inflation pressure with a constant vertical load value resulted in an increase in the contact area, but this trend was not observed for the bias-ply tire. The bias-ply tire generated a footprint of a smaller width and length, but a greater depth and contact area, than the radial tire. This indicates that the depth of a tire footprint largely determines the contact area. It was noted that a comparable contact area of the tires could be achieved for different combinations of inflation pressure and vertical load, which means that an ideal combination could reduce the depth of the tire footprint. This is very important information for agricultural practice, because further research will make it possible to use tires on a field while avoiding soil environment degradation.

**Author Contributions:** Conceptualization, J.C. and W.P.; methodology, J.C.; software, M.B. and W.P.; validation, M.B. and A.M.; formal analysis, J.C. and K.L.; investigation, W.P., M.B. and A.M.; resources, W.P.; data curation, M.B.; writing—original draft preparation, W.P. and M.B.; writing—review and editing, M.B.; visualization, W.P.; supervision, J.C.; project administration, J.C. and K.L.; funding acquisition, J.C. and K.L. All authors have read and agreed to the published version of the manuscript.

**Funding:** The article processing charge was financed by Wroclaw University of Environmental and Life Sciences.

**Institutional Review Board Statement:** Not applicable.

**Informed Consent Statement:** Not applicable.

**Data Availability Statement:** Not applicable.

**Conflicts of Interest:** The authors declare no conflict of interest. The funders had no role in the design of the study; in the collection, analyses, or interpretation of data; in the writing of the manuscript; or in the decision to publish the results.

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
