# Peer review of "Evaluation of Tire Footprint in Soil Using an Innovative 3D Scanning Method"

_agriculture, doi:10.3390/agriculture13030514_

Round 1
Reviewer 1 Report
The research is topical and is related to the issue of soil quality preservation.
1. In essence, the depth of the tread indentation depends on the area of the treads themselves. How does the type of tread itself affect tire tread depth? How does this factor into your research?
2. When using mathematical methods, their results are not affected by the specific application. Are there any physical, practical conditions for using the logarithmic, variable inverse function in equations (1) and (3) for data description?
3. The transfer of any mathematical expression to reality requires limits of application. For what values of p and G are the relations (1) and (3) applicable
Author Response
Dear Reviewer,
Thank you for the reviewing process and all your comments. The answers to them were described in the sentences below:
1. In essence, the depth of the tread indentation depends on the area of the treads themselves. How does the type of tread itself affect tire tread depth? How does this factor into your research?
Answer: We analysed one type of tread pattern (it was the same it both tyres). In this research the tread pattern was not a factor. However, in further research we could analyse the relationships between technical tyres parametres and soil deformation.
2. When using mathematical methods, their results are not affected by the specific application. Are there any physical, practical conditions for using the logarithmic, variable inverse function in equations (1) and (3) for data description?
Answer: The mathematical modelling process was conducted just for conditions of our research. The data obtained from measurements were using in statistical software- based on them the mathematical functions were created. The models were chosen based on the highest values of coefficient of determination (R2). Then, the mathematical verification of models were conducted using f-Snedecor function. The date used to verification od models came from other measurements, than the data used to mathematical modelling process.
3. The transfer of any mathematical expression to reality requires limits of application. For what values of p and G are the relations (1) and (3) applicable
Answer: Created models can be used in limitation range of inflation pressure from 0.8 to 2.4 bar. The limitation range for the second factor (vertical load) was 7.8 kN to 23.5 kN.
We would like to thank you once again for the review process and all comments and suggestions.
Sincerely
Authors
Reviewer 2 Report
The paper describes the experimental test of two different tyres footprints, and the length, width, depth, and contact-area under different inflation pressure and vertical load are obtained and analyzed. The mathematical models were formulated to describe contact area. Generally, the paper's topic is interesting and clearly shows a great amount of work. However, there are some questions to be solved.
1. The tire footprint test was conducted in laboratory conditions, and the test bench schematic design is shown in Figure 1. What I wonder is why the physical field test does not appear in the paper. Based on the loss of the field test, the test results of the footprint are the least reliable,
2. Is the footprint of figure 3 the radial tire or bias-ply tire? the tire footprint test results are in a small number, and the actual field test is also not provided, thus, the footprint parameters are also still questionable.
3. How do you decide the depth of the footprint h? or, what criteria is selected to obtain the depth?
Author Response
Dear Reviewer,
Thank you for the reviewing process and all your comments. According to your suggestion, the manuscript has undergone English language editing by MDPI. The answers to your questions were described in the sentences below:
1. The tire footprint test was conducted in laboratory conditions, and the test bench schematic design is shown in Figure 1. What I wonder is why the physical field test does not appear in the paper. Based on the loss of the field test, the test results of the footprint are the least reliable.
Answer: In thise case, the article doesn’t include the results of field test, because it is one of the papers from whole cycle of research. The first stage in this cycle was evaluation of utility of our method to determine the deformation of tyres on non-deformable surface. Next phase of the cycle concerned evaluation of tyre deflection in laboratory conditions (on non-deformable plate). Reviewed paper is third stage of the described cycle- so it described deformation the soil under acting tyres in static condisins, in laboratory. Final stage of the cycle will concern description of result obtained from field tests. In whole cycle the same tyres were used, experiment parameters were also the same. Both in laboratory and field tests the same type of soil was used. Comparision the results from laboratory part and field part should give the answer to the question about possibility of using proposed method in laboratory conditions to determine basic parametres related to soil compaction. In turn, it allows to avoid to conduct expensive and time-consuming fild experiment.
2. Is the footprint of figure 3 the radial tire or bias-ply tire? the tire footprint test results are in a small number, and the actual field test is also not provided, thus, the footprint parameters are also still questionable.
Answer: Figure 3 presents footprint of radial tyre. This is only an example to shows points of (lenght and width of tire fottpirnt), which were always measured in the same place of references. We believe, results obtained from laboratory part could be used to predict of parametres in real field conditions, however the verification of it is needed, what will be presented in the next paper.
3. How do you decide the depth of the footprint h? or, what criteria is selected to obtain the depth?
Answer: To measure the depth of tyre footprint the vertical cross- section across the middle of the footprint was created. The cross- section contain artificial line, wchich was a reference point to the ground outside the footprint. The height of cross section was equal to tyre footprint depth. Important criteria by measure tyre footprint depth was measurement the paramenter always in the same place (in the middle of cross scetion).
We would like to thank you once again for the review process and all comments and suggestions.
Sincerely
Authors
Reviewer 3 Report
Dear Authors,
in order to improve your article, I recommend providing answers to the following questions in the article:
• Was the wheel just placed in the soil sample, or did the wheel rotate?
• Did the wheel act on the ground as a driving or braking force? Would the driving force or braking force affect the pressure distribution in the soil?
• Do the authors assume that the greatest soil compaction in the vertical direction is in the center of the tire? Can't there be a higher load on the sides of the tire in some cases?
• In the article, I recommend to state more clearly the scientific and practical contribution of the article. What is innovative about your article?
• For a scientific article, I recommend using contemporary literature. I do not recommend using literature from 1976 and similar in scientific literature.
Sincerely
Reviewer
Author Response
Dear Rewiever,
Thank you for your comments and suggestions. According to your suggestion, the manuscript has undergone English language editing by MDPI.
Anwers for your qustions and recommendations written bellow ale included in article and marked in the text:
• Was the wheel just placed in the soil sample, or did the wheel rotate?
• Did the wheel act on the ground as a driving or braking force? Would the driving force or braking force affect the pressure distribution in the soil?
• In the article, I recommend to state more clearly the scientific and practical contribution of the article. What is innovative about your article?
• For a scientific article, I recommend using contemporary literature. I do not recommend using literature from 1976 and similar in scientific literature.
Answer for question: "• Do the authors assume that the greatest soil compaction in the vertical direction is in the center of the tire? Can't there be a higher load on the sides of the tire in some cases?" is not included in the text and is written bellow:
Answer: In presented paper there were described the results concerning only the footprint in the soil. We didn’t analyse the distribution of pressures in the layers below the footprint. This issues will be analyse in the further parts of research cycle.
We would like to thank you once again for the review process and all comments and suggestions.
Sincerely
Authors
Round 2
Reviewer 1 Report
The article is interesting, especially for agricultural practitioners.
Research on relationships between technical tire parameters and soil deformation might be interesting.
Reviewer 2 Report
NOTHING
Round 3
Reviewer 2 Report
nothing